# *CDH1* (*E-cadherin*) Gene Methylation in Human Breast Cancer: Critical Appraisal of a Long and Twisted Story

**DOI:** 10.3390/cancers14184377

**Published:** 2022-09-08

**Authors:** Lara Bücker, Ulrich Lehmann

**Affiliations:** Institute of Pathology, Medical School Hannover, D-30623 Hannover, Germany

**Keywords:** DNA methylation, CDH1, *E-cadherin*, breast cancer, invasive lobular breast cancer (ILBC)

## Abstract

**Simple Summary:**

Genes can be inactivated by specific modifications of DNA bases, most often by adding a methyl group to the DNA base cytosine if it is followed by guanosine (CG methylation). This modification prevents gene expression and has been reported for many different genes in nearly all types of cancer. A prominent example is the gene *CDH1*, which encodes the cell-adhesion molecule *E-cadherin*. This is an important player in the spreading of tumor cells within the body (metastasis). Particularly in human breast cancer, many different research groups have studied the inactivation of the CDH1 gene via DNA methylation using various methods. Over the last 20 years, different, in part, even contradicting results have been published for the CDH1 gene in breast cancer. This review summarizes the most important publications and explains the bewildering heterogeneity of results through careful analysis of the methods which have been used.

**Abstract:**

Epigenetic inactivation of a tumor suppressor gene by aberrant DNA methylation is a well-established defect in human tumor cells, complementing genetic inactivation by mutation (germline or somatic). In human breast cancer, aberrant gene methylation has diagnostic, prognostic, and predictive potential. A prominent example is the hypermethylation of the *CDH1* gene, encoding the adhesion protein E-Cadherin (“epithelial cadherin”). In numerous publications, it is reported as frequently affected by gene methylation in human breast cancer. However, over more than two decades of research, contradictory results concerning *CDH1* gene methylation in human breast cancer accumulated. Therefore, we review the available evidence for and against the role of DNA methylation of the *CDH1* gene in human breast cancer and discuss in detail the methodological reasons for conflicting results, which are of general importance for the analysis of aberrant DNA methylation in human cancer specimens. Since the loss of *E-cadherin* protein expression is a hallmark of invasive lobular breast cancer (ILBC), special attention is paid to *CDH1* gene methylation as a potential mechanism for loss of expression in this special subtype of human breast cancer. Proper understanding of the methodological basis is of utmost importance for the correct interpretation of results supposed to demonstrate the presence and clinical relevance of aberrant DNA methylation in cancer specimens.

## 1. Introduction

The *CDH1* gene encodes the transmembrane protein *E-cadherin* that mediates Ca^2+^-dependent cell–cell adhesion. *E-cadherin* belongs to the classical cadherins family, also known as type 1 cadherins [1]. The name was given because of its typical expression in epithelial cells [2]. *E-cadherin* consists of five “tandemly repeated” extracellular domains, EC1 to EC5 [1,3], a transmembrane domain, and a cytoplasmic domain [1]. Each extracellular domain has three binding sites for Ca^2+^, which is important for stabilizing cell–cell adhesion [3,4]. The intracellular domain binds to various signaling molecules such as β-catenin. β-catenin indirectly connects *E-cadherin* to the actin cytoskeleton via α-catenin, granting greater robustness to cadherin adhesion [5]. Loss of this cell–cell adhesion can have a major impact by causing or enabling tumor cells to migrate, leading to cancer progression and metastasis [6,7].

## 2. Genetic or Epigenetic in Activation

Loss of *E-cadherin* is involved in the development and progression of several human malignancies [8]. The *CDH1* gene can be inactivated by genetic and epigenetic mechanisms. Germline mutations are responsible for Hereditary Diffuse Gastric Cancer (HDGC) [9], a genetic syndrome linked to a lifetime risk of gastric cancer of 30–60%. Carriers also have a 40–50% lifetime risk of invasive lobular breast cancer (ILBC) [10].

Epigenetic inactivation by aberrant DNA methylation has been reported especially for gastric and breast cancer but also for bladder and colon cancer. One decade ago, these data reached the level of (presumably) secured textbook knowledge (see Table 7.2 in Robert Weinbergs widely acclaimed textbook about the biology of cancer, [11]).

Alternatively, the expression of the *CDH1* gene can be downregulated by transcriptional repression mediated by the transcription factors SNAIL, SLUG, ZEB1, and ZEB2/SIP1 [8,12].

Since inactivation of the *CDH1* gene is an important step in the metastasis-promoting process of epithelial–mesenchymal transition (EMT, [7]), elucidation of the involved mechanisms is of importance for a better understanding of the development and progression of human breast cancer [13].

## 3. DNA Methylation

The most widespread and abundant form of DNA methylation across many different eukaryotic species is the modification of carbon atom no. 5 of the DNA base cytosine (cytosine-5 methylation or 5-methyl cytosine) in the context of the dinucleotide CG (often referred to as CpG methylation where “p” stands for the phosphodiester bond of the DNA backbone) [14].

The role of 5-methyl cytosine in eukaryotic gene expression was hypothesized 50 years ago (see ref. [15] for an early overview and relevant primary references from the early days of DNA methylation research).

Cytosine methylation leads to changes in chromatin structure and gene expression by (1) direct interference with transcription factor binding, (2) recruitment of methylcytosine binding proteins (e.g., MeCP2), and (3) the induction of a closed chromatin structure (i.e., heterochromatization) [16,17].

DNA methylation is involved in (1) the inactivation of the second X-chromosome in female cells, (2) the establishment and maintenance of mono-allelic gene expression of imprinted genes, (3) the repression of potentially autonomous DNA elements within our genome, and (4) the maintenance of chromosomal integrity by preventing illegitimate recombination between homologous regions in the genome [18,19].

Under most circumstances, DNA hypermethylation leads to the inactivation of gene expression. However, heterochromatization of repressive elements by hypermethylation or prevention of the binding of a repressor to its specific binding site can indirectly lead to an activation of gene expression [20]. Therefore, the simple textbook formula “DNA methylation = gene repression” is not always valid.

The role of DNA methylation “in the generation of tumor heterogeneity and progression” was put forward 40 years ago (see ref. [21] for an early review and many primary references from that time). The role of aberrant DNA cytosine methylation in the development and progression of human tumors is now well-established [22,23,24]. The very first example of the epigenetic inactivation of a *bona fide* tumor suppressor gene was discovered by the group of Bernhard Horsthemke (Department of Human Genetics, University Hospital Essen, Germany). In a careful study of 21 retinoblastoma patients, they demonstrated the hypermethylation of the *RB1* gene in sporadic unilateral retinoblastoma [25], reviewed in ref. [26]. Many tumor suppressor genes were added to this list in the following years (see above-cited reviews).

## 4. Detection of DNA Methylation

### 4.1. Bisulfite-Based

The vast majority of DNA methylation studies are based on the treatment of genomic DNA with highly concentrated bisulfite, which converts the epigenetic modification of the methyl group at carbon atom no. 5 of the DNA base cytosine into a difference in the primary sequence [27,28]. Subsequently, it can be analyzed in qualitative and quantitative terms employing a wide range of molecular techniques developed for the detection of single nucleotide variants [29,30]. The huge advantage of the bisulfite conversion for DNA methylation studies is the fact that after treatment with bisulfite, various techniques can be used which are familiar to most researchers and are available in nearly all laboratories. The disadvantage is that the harsh chemical conditions lead to extensive fragmentation of the genomic DNA and the creation of abasic sites, reducing the sensitivity of the approach and increasing the danger of sequencing artefacts [31,32]. After bisulfite treatment, methylated DNA is more similar to untreated DNA than unmethylated DNA. Incomplete bisulfite treatment can result in false-positive methylation calls. Therefore, controlling for the completeness of the bisulfite treatment is of uppermost importance. Unfortunately, two of the most popular methods in the field, i.e., MSP and MethyLight (see below), do not allow for proper control of the completeness of the bisulfite treatment.

Undisputedly, the most frequently used method for the detection of DNA methylation, especially for the analysis of primary human samples, is methylation-specific PCR, MSP [33], a variation of allele-specific PCR, with one primer pair amplifying methylated DNA and another primer pair amplifying unmethylated DNA after treatment of the DNA with bisulfite. Methylation-specific PCR leads to the democratization of DNA methylation analysis because it is easy to perform, quite cheap, and does not require special laboratory equipment. In addition, only comparatively small amounts of genomic DNA are required. This method undoubtedly pushed the field of DNA methylation research forward and deserves appropriate credit for this achievement. However, MSP is hampered by major drawbacks and contributed (and still contributes!) to a substantial amount of false-positive claims and thereby flawed publications. The most important problem is the overestimation of weak PCR products as proof of biologically and clinically relevant DNA hypermethylation. Since end-point PCR is not able to provide quantitative information, spurious DNA methylation is often interpreted as “hypermethylation” of the gene of interest. It is often overlooked that “hypermethylation” is a quantitative concept (= “more methylated than”), which requires a quantitative method. Additionally, cross-reactivity of M-primers with unmethylated DNA leads to false-positive results. Since MSP cannot control for completeness of bisulfite treatment, this source of false-positive methylation results cannot be controlled (and contaminates the literature to an unknown extent). Methylation-specific PCR also relies on homogeneous methylation patterns and is not able to dissect heterogeneous methylation patterns correctly.

Another straightforward, easy-to-perform and cheap method from the early days of DNA methylation research in the cancer field is COBRA, combined bisulfite treatment and restriction analysis [34]. This approach is somewhat limited by the presence (or absence) of suitable restriction sites within the region of interest (created or destroyed by the C-T-conversion). The relatively low sensitivity of this method is compensated by inherent robustness against false-positive results. However, COBRA is rarely used for the detection of *CDH1* gene methylation in human breast cancer.

The adaptation of real-time PCR protocols for the analysis of DNA methylation represents an important methodological innovation in the field. The so-called “MethyLight” assay (and related methodologies, [35,36]) is a widely used quantitative method which allows for sensitive and specific detection of DNA methylation. However, confirmation of 100% efficiency of the bisulfite treatment is not directly possible, and many quantification algorithms rest on the amplification of a single reference locus (often *ACTB*, see ref. [35]), making this approach susceptible to systematic bias in computing methylation levels [37].

High-resolution melting is a variation of methylation-specific real-time PCR [38], which also provides quantitative information. It is a fast, simple, and cheap technique which can be performed in most laboratories. A limitation of this approach is the inability to dissect and quantify heterogeneous methylation patterns.

Sanger sequencing-based methods have been used for DNA methylation studies from early on, either by direct sequencing of bisulfite-treated DNA or by sequencing of individual plasmids after cloning of PCR products into appropriate vectors [39]. Direct sequencing is the most straightforward, easy to perform, and is able to resolve heterogeneous methylation patterns; however, it only provides semi-quantitative information at best. If a sufficient number of clones have been sequenced already, old-fashioned Sanger bisulfite sequencing is able to provide high-resolution quantitative pictures of methylated regions of the human genome, albeit with low throughput. Nowadays, the sequencing part of this approach has been overtaken by various NGS-based protocols [40], combining exact high-resolution quantification with high throughput.

Pyrosequencing offers the possibility of high-resolution quantitative detection of DNA methylation [41,42]. In contrast to all direct PCR methods, it provides quantitative information for each CG site under study and is, therefore, capable of resolving complex heterogeneous methylation patterns. It also allows for control of bisulfite conversion efficiency.

All above-described methods analyzing bisulfite-treated DNA, which include a PCR amplification step (or even two or three), face the problem of potential PCR bias, i.e., the preferential amplification of either (T-rich) unmethylated DNA or of (CG-rich) methylated DNA, thereby distorting the DNA methylation results. Unfortunately, the vast majority of publications do not present data documenting unbiased amplification of methylated and unmethylated DNA [43,44].

Bead-array-based analysis of CpG methylation using the 450 k array from Illumina (or its successor, the EPIC array, covering approx. 850,000 CpG sites) became the industry standard for comprehensive profiling of the human genome in recent years, contributing to real progress in research and diagnostics [45,46,47,48]. The huge advantage is the robustness of the methodology and the standardization of DNA methylation profiles all over the world. Numerous software packages for the 450 k and the EPIC array are freely available, making data evaluation transparent and reproducible [49].

The representation of the human genome on the 450 k or the EPIC array is still uneven with a bias towards “cancer genes”, reflecting the product history from the Golden Gate assay (covering approx. 1500 CpG sites [50]) to the EPIC array (with 850,000 CpG sites covered [46]). Genomic regions that are represented by only a few probes or not at all on these arrays are “forgotten” in the literature.

### 4.2. Bisulfite-Free

Affinity-enrichment of methylated DNA either by anti-methyl cytosine-antibodies or methyl cytosine binding proteins (MBD) [51,52] has the advantage of circumventing all problems with bisulfite artifacts and bisulfite-induced DNA loss. For the analysis of the enriched DNA fraction, which represents methylated loci, one can use quantitative PCR (for analysis of only a few loci), array-hybridization, or next-generation sequencing [53]. However, anti-methyl cytosine antibodies exhibit limited specificity and sensitivity.

Nanopore sequencing offers, in principle, the opportunity to detect all known base modifications during the process of DNA sequencing without the requirement of any treatment of the DNA before it is sequenced [54,55]. DNA methylation analysis via nanopore sequencing is still in quite an early phase of development and the analysis of primary patient material, which often provides only limited amounts of genomic DNA, is far from well-established. Figure 1 provides an overview of the most frequently used methods.

## 5. *CDH1* Gene Methylation

The aim of this review is not to compile each and every paper ever published on *CDH1* gene methylation in human breast cancer. Instead, a critical overview of the history of *CDH1* gene methylation in human breast cancer with a strong focus on the methodology used is provided, which cites all relevant studies published so far with a thorough discussion of the methodological reasons for contradictory results.

The first report to state that the *CDH1* gene might be affected by aberrant DNA methylation in human carcinoma cells was published in 1995 [63]. By Southern blot-analysis of 11 human carcinoma-derived cell lines, Yoshiura et al. collected some evidence that *Hpa II* recognition sites around the promotor of the *CDH1* gene are methylated in those cell lines that showed no *E-cadherin* mRNA expression. The problem with this approach is that the absence of a signal (= band in a gel or blot) is taken as evidence of DNA methylation. Therefore, any inhibitory effect preventing the proper activity of the restriction enzyme leads to false-positive results. At that time, it was obviously acceptable to talk about “human carcinomas” if one was analyzing only carcinoma-derived cell lines, some already decades old. Yoshiura et al. analyzed only MCF7 as a breast cancer-derived cell line, which showed no evidence of gene methylation in their analyses.

In the following years, numerous publications reported *CDH1* gene methylation in various malignancies [64,65]. The vast majority of these studies employed non-quantitative MSP, which tends to overestimate the frequency and extent of gene methylation. Figure 2 illustrates the location of the CpG sites analyzed by various methods.

In addition, one has to keep in mind that DNA methylation patterns in established cell lines are, beyond any doubt, very distinct from DNA methylation patterns in primary tissue samples. This was demonstrated during the last 30 years by many independent research groups for many different cell lines and primary tissue specimens: see refs. [66,67,68,69] and the references therein. Therefore, most statements about DNA methylation of a particular gene in human cancer that are solely based on cell line studies are of very limited use. Cell line studies of DNA methylation aberrations can only form the hypothesis-formulating starting point for the analyses of gene methylation in primary patient samples.

## 6. In Breast Cancer

Graff et al. [70] were the first to show *CDH1* gene methylation in primary breast cancer specimens. By Southern Blot analysis, the authors examined 12 primary breast cancer specimens altogether (including two breast cancer metastases). In this type of assay, the absence of restriction enzyme activity is interpreted as the presence of DNA methylation: inhibition of enzymatic activity by methylation of cytosine within the recognition site leads to the appearance of larger, non-digested fragments. Depending on the specific enzyme used, up to 7 out of 10 samples displayed evidence of DNA methylation in this study. Unfortunately, the authors of this pioneering study did not attempt to quantify the band intensity. From a visual inspection of the figures provided (Figure 2, Panel C and D in [70]), it is obvious that only a very small amount of genomic DNA is not cut by the various restriction enzymes, indicating very weak DNA methylation. Small amounts of impurities and/or suboptimal reaction conditions for the restriction enzymes could be responsible for the appearance of these very weak larger (uncut) bands, which are interpreted as evidence for the presence of DNA methylation. In a follow-up study regarding *CDH1* gene methylation in human breast cancer, Graff et al. analyzed only well-established cell lines and eight primary normal breast tissue specimens, but no additional primary human breast carcinoma samples were used [71].

In the frequently cited overview “A Gene Hypermethylation Profile of Human Cancer” by Esteller and colleagues in 2001 [72], it is stated that the *CDH1* gene is “hypermethylated” in 42% (37/88) of human breast cancer specimens. Unfortunately, it is not possible to figure out which breast cancer samples (only ductal-invasive or also lobular-invasive? Fresh-frozen or FFPE?) have been analyzed using which method. Most probably, methylation-specific PCR was used.

A couple of years later, Shinozaki et al. [73] analyzed 151 primary breast tumors and 29 sentinel lymph node metastases using MSP with self-designed primers. The PCR products were analyzed using capillary electrophoresis. Unfortunately, no primary data for *CDH1* gene methylation are presented in this paper. The authors claimed to have found “*CDH1* methylation” in 80 primary tumor samples (53%) and 26 sentinel lymph node metastases (90%).

Caldeira et al. [74] employed a nested MSP approach which is even more prone to false-positive results and found *CDH1* gene methylation in 56/76 of cases (74%). These authors also described CDH1 hypermethylation in six cases which were clearly positive for *E-cadherin* expression in the tumor cells, demonstrated by unequivocal membranous *E-cadherin* staining (similar to Droufakou et al. [75], see below).

Toyooka et al. [76] developed one of the first quantitative assays for the measurement of *CDH1* gene methylation, employing real-time PCR and the unmethylated *MYOD1* locus as input control. These authors adjusted the threshold of the quantitative assay in order to obtain the best concordance with the qualitative end-point MSP instead of developing stringent criteria for scoring a sample as “*CDH1* gene methylated”. Thereby, 25 out of 56 (45%) specimens were designated as “*CDH1* gene methylated”.

Sebova et al. [77] used quantitative multiplex methylation-specific PCR (QM-MSP, [78]) for the analysis of *CDH1* gene methylation in 92 archival (i.e., formalin-fixed and paraffin-embedded) human breast cancer samples. Twenty (22%) showed evidence for *CDH1* gene methylation, and two out of seven lobular invasive breast cancer specimens displayed *CDH1* gene methylation. Swift-Scanlan et al. [79], 2011, also employed the same real time-PCR-based approach; however, they defined 5% as the threshold for scoring a sample as “methylated”. Out of 99 archival primary breast cancer specimens, 2 were scored as “*CDH1* methylated” (2%). The maximum methylation level of 21% for the *CDH1* gene (see Table 3 in Swift-Scanlan et al. 2011) was well below the maximum methylation levels reported for the nine other cancer-related genes analyzed in this study (84 to 96%).

These results clearly indicate that *CDH1* gene methylation might be much less frequent and much less pronounced than reported before by several groups using non-quantitative methods. Additionally, with the very low threshold of 0.5% for scoring a sample as “methylated”, the use of a quantitative approach substantially reduced the fraction of primary patient samples considered as “*CDH1* gene methylation positive” in comparison to earlier studies using non-quantitative MSP.

Feng et al. [80] used pyrosequencing instead of quantitative, real-time PCR-based MSP. These authors analyzed 90 breast cancer specimens altogether. For each case, tumor and adjacent normal tissue were available. However, the *CDH1* gene was analyzed in only 34 samples of this cohort. In all tumor and adjacent normal samples, only background methylation of approximately 4% could be detected, confirming the results obtained by quantitative MSP.

However, even years after these publications, research groups are still employing non-quantitative MSP and report mostly exaggerated numbers for a fraction of primary breast cancer specimens displaying “CDH1 gene methylation” above biological and technical background. For example, Liu et al. [81] analyzed 137 primary breast tumors and 13 metastases reporting “CDH1 methylation” in 41% of cases but only a marginal reduction in gene expression.

Naghitorabi et al. [82] employed D-HRMA and found “CDH1 methylation” in all breast cancer tissue samples under study (*n* = 98, FFPE) as well as in 90% of the normal breast tissue samples (*n* = 10, fresh-frozen), raising serious questions about the specificity of their approach.

Employing a very sophisticated single-cell approach on a limited number of samples, Pixberg et al. [83] found only marginal methylation levels for the *CDH1* gene in carcinoma cells below the levels found in CD45+ leukocytes. These data derived from the analysis of single cells from 11 patients are in line with larger studies employing quantitative methodology for the analysis of bulk tumor samples, all indicating that *CDH1* gene methylation is not frequent in breast cancer and is very often found at a comparatively low level, questioning the functional relevance.

In a study of 60 male breast cancer specimens with a familial background, Deb et al. [84] also found no evidence of *CDH1* gene methylation in this special subgroup of human breast cancer using HRM. Male breast cancer makes up approximately 1% of all human breast cancer cases [85].

In a large study comprising 855 breast cancer cases, McCullough et al. [86] found *CDH1* gene methylation in 5–10% of cases, depending on the subgroup. They employed a quantitative methodology, albeit with quite a low threshold of 4% PMR. This percentage point is not directly comparable with the DNA methylation levels measured by pyrosequencing, NGS, or bead arrays (all three are also expressed as a percentage). Nevertheless, this study confirms the finding that the use of quantitative methodology dramatically reduces the number of specimens scored as “*CDH1* gene methylation positive”.

In the re-analysis of the multi-omics TCGA data set (*n* = 981), Sivadas A et al. [87] identified several differentially methylated loci in human breast cancer; however, they did not report *CDH1* as differentially methylated.

## 7. *CDH1* Gene Methylation in Lobular Breast Cancer

Loss of *E-cadherin* protein expression is a defining hallmark of lobular breast cancer, a histological subtype which comprises 10–15% of all primary human breast cancer cases [88].

The first study addressing epigenetic inactivation of the *CDH1* gene as a potential molecular mechanism for the loss of *E-cadherin* protein expression in the context of human lobular breast cancer appeared in 2001 [75]. Droufakou et al. studied 22 ILCs employing methylation-specific PCR and found that 17/22 (77%) of the specimens “had methylation of the *CDH1* promotor, including 11/12 (91%) of *E-cadherin*-negative tumours.” This interesting statement from the abstract hinted (unwillingly?) at the lack of specificity of the assay used because it implies that six tumours, which were *E-cadherin*-positive, showed promotor hypermethylation. Altogether, 17 specimens showed *CDH1* gene methylation, 11 of these were *E-cadherin*-negative, leaving 6 specimens which are *E-cadherin*-positive and *CDH1* gene methylation-positive. This contradiction is not explained in any way by these authors.

Two years later, Sarrio et al. [89] published a larger study analyzing 46 invasive lobular breast cancer specimens for the presence of *CDH1* gene methylation using the same approach as Droufakou et al. (MSP employing primers described first by Herman et al. in 1996 [33]). In this second study, 19/46 (41%) showed *CDH1* gene methylation. The authors stressed that only 20% of CDH1 expression-positive samples also displayed *CDH1* gene methylation compared to 60% in Droufakou et al., taken by Sarrion et al. as an indicator of improved assay quality. Again, these authors did not elaborate on the contradiction of *CDH1* gene methylation and *E-cadherin* protein expression in the same breast carcinoma specimen.

In the above-mentioned study from Shinozaki et al. [73], 33 lobular breast cancer specimens were included, and 12 (36%) displayed evidence of “*CDH1* methylation”. The study from Liu et al. [81] comprised 31 lobular breast cancer specimens, 8 (25%) with “*CDH1* methylation” (see Table 1 for a summary of data).

Zou et al. analyzed 14 primary invasive lobular breast cancer specimens, 13 of them displaying *CDH1* gene methylation employing non-quantitative MSP [91]. All ALH/LCIS (*n* = 14) and non-neoplastic epithelium samples (*n* = 8) in this study showed methylation signals employing non-quantitative MSP, leading the authors to the conclusion that *CDH1* gene methylation is an early event in lobular breast cancer development. Attempting to demonstrate the specificity of their approach, these authors sequenced the products of the PCRs using the primers amplifying methylated DNA (“M-PCR products”) for five specimens, all showing nearly complete methylation (Figure 2 in Zou et al., 2009). However, this does not solve the problem of exaggerating spurious methylation events using a non-quantitative methodology.

In a groundbreaking publication, Ciriello et al. presented within the framework of the TCGA consortium the most comprehensive molecular characterization of invasive lobular breast cancer, analyzing a cohort of 127 ILBCs employing several comprehensive profiling technologies [92]. Citing earlier studies regarding *CDH1* gene methylation in lobular breast cancer [70,91], these authors reanalyzed the results for the six 450 k array probes interrogating *CDH1* gene methylation status (Figure S2D,E in Ciriello et al. 2015). In addition, in five samples, the *CDH1* locus was manually reanalyzed using deep genome bisulfite sequencing. Both approaches provided no evidence for *CDH1* gene methylation within the promotor and exon 1 in human lobular breast cancer. Ciriello et al. point to the widespread use of the non-quantitative, very sensitive MSP method as a potential source of false-positive results and ask for further investigations to clarify these contradictory results.

In the most recent original study of *CDH1* gene methylation in ILBC, Alexander et al. analyzed 18 classical invasive lobular breast cancers using the 850 k array (also called EPIC array) and clearly stated: “In agreement with the recent TCGA study, we did not identify promoter methylation of *CDH1* in any of the tumors” [93].

## 8. DNA Methylation Is Cell-Type Specific

In 2004, Lombaerts et al. [90] identified an important confounding factor for the analysis of *CDH1* gene methylation in primary tissue samples: The partial methylation of the *CDH1* gene in leukocytes, which are found in nearly all tissue specimens, can lead to false-positive results if MSP or related methodologies are used. This was also described many years ago for several other potentially interesting targets of aberrant DNA methylation in human breast cancer (e.g., *14-3-3 sigma*, [94] or *HOXA5* [95]). Unfortunately, these insights are overlooked by many research groups. This means that the published *CDH1* methylation data are contaminated to an unknown extent by false-positive calls due to leukocyte infiltrates in primary breast cancer specimens. This is a challenge, especially in lobular breast cancer, which is characterized by a discohesive growth pattern (“Indian file”), typically infiltrating as single cells, not forming a compact tumor mass comprising only little non-neoplastic cells [96].

Even after the publication of Lombaerts et al. [90], studies on *CHD1* gene methylation in primary patient samples appeared, which employed non-quantitative non-cell type-specific methodology (primarily MSP analysis of bulk tumor samples).

## 9. The Problem with Reviews

The value of many reviews published during the last decade regarding gene methylation in breast cancer is reduced by the fact that data reported in PubMed are taken at face value and compiled without any critical evaluation and discussion of the methodology in use. Thereby, false-positive findings will be confirmed as alleged true-positive results and passed down to the next generation of researchers. Examples are the reviews from Kristensen et al. [97], Huang et al. [98], Davalos et al. [99], and Ruijter et al. [100]. Occasionally, even clearly wrong papers are cited, not dealing with *CDH1* gene methylation at all (e.g., ref. 11 and 30 in ref. [97]).

In a very recent review regarding ILBC ([101], submitted in December 2021), it is clearly stated that “epigenetic silencing by hypermethylation of the *CDH1* gene promoter did not appear to be associated with *E-cadherin* downregulation”, citing Ciriello et al. 2015 [92] as a reference, indicating that after 25 years of research the vanishing importance of *CDH1* gene methylation in ILBC is documented now also in reviews and not only in primary studies.

In conclusion, one has to say that the literature on *CDH1* gene methylation in human breast cancer is contaminated by numerous false-positive results due to the use of non-quantitative methods in a non-cell type-specific manner resulting in the overestimation of spurious methylation signals (in part coming from non-cancer cells). Over the years, with the increasing use of high-resolution quantitative methods for the detection of DNA methylation, the fraction of primary human breast cancer samples reported displaying *CDH1* gene methylation continuously declined. By analyzing the increasing number of samples and employing a more sophisticated quantitative high-resolution detection methodology for invasive lobular breast cancer, the most frequent special type of breast cancer [96,102], the frequency and the level of *CDH1* gene methylation approached nearly zero.

Many contradictory results concerning the association of *CDH1* gene methylation with histological and/or clinical parameters will be resolved in the future if standardized quantitative methods and stringent threshold settings are employed.

## 10. Wider Implications

The critical review of DNA methylation data presented in this manuscript should lead to re-evaluation of other genes reported to be affected by gene methylation with a strong focus on the methodology employed. A prominent candidate for such re-evaluation is the tumor suppressor gene *BRCA1* [103]. The failure to establish *BRCA1* gene methylation as a predictive marker for response to PARP inhibitor therapy might be due to methodological shortcomings [104]. In the past, patient samples with only spurious methylation of the *BRCA1* gene might have been erroneously scored as “methylated”. The resulting dilution of truly *BRCA1* methylated patient samples (with epigenetic inactivation of this crucial DNA repair gene within the tumor cells) in a large group of false-positive samples might have masked the association between *BRCA1* gene methylation and clinical response to PARP inhibitor therapy. Recent data indicate that thorough quantitative analysis can establish *BRCA1* methylation as a predictive marker for response to PARP inhibitor therapy [105] and might even explain the development of resistance in selected cases [106].

Rephrasing the title of the stimulating commentary of van Vlodrop et al., which appeared more than 10 years ago in the journal *Clinical Cancer Research* [107], one could conclude it should be entitled: Analysis of promotor CpG Island hypermethylation in Cancer: Method, Method, Method!

## Figures and Tables

**Figure 1 cancers-14-04377-f001:**
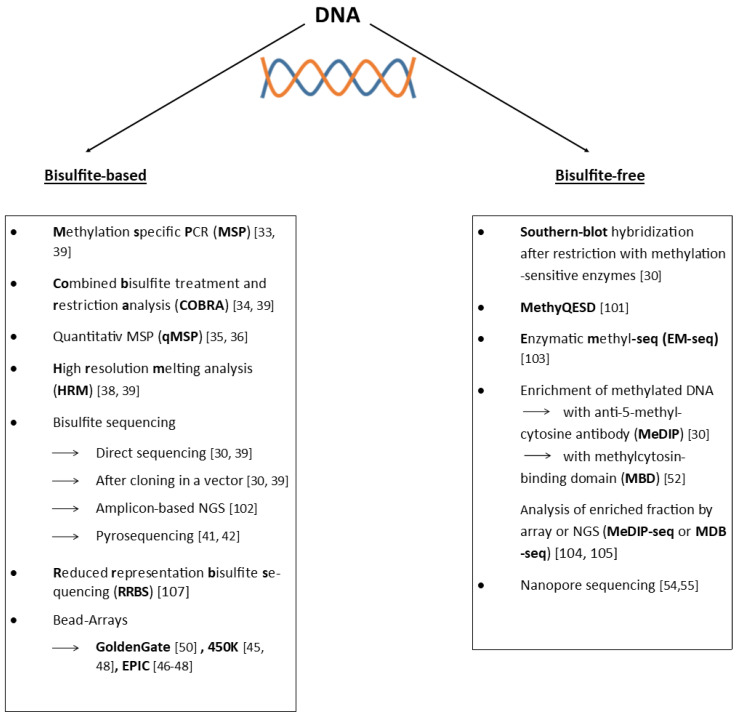
Conceptual overview of the most frequently used techniques used for studying DNA methylation [30,33,34,35,36,38,39,41,42,45,46,47,48,52,54,55,56,57,58,59,60,61]. More in-depth information about the methodologies and useful further references can be found in refs. [30,62].

**Figure 2 cancers-14-04377-f002:**
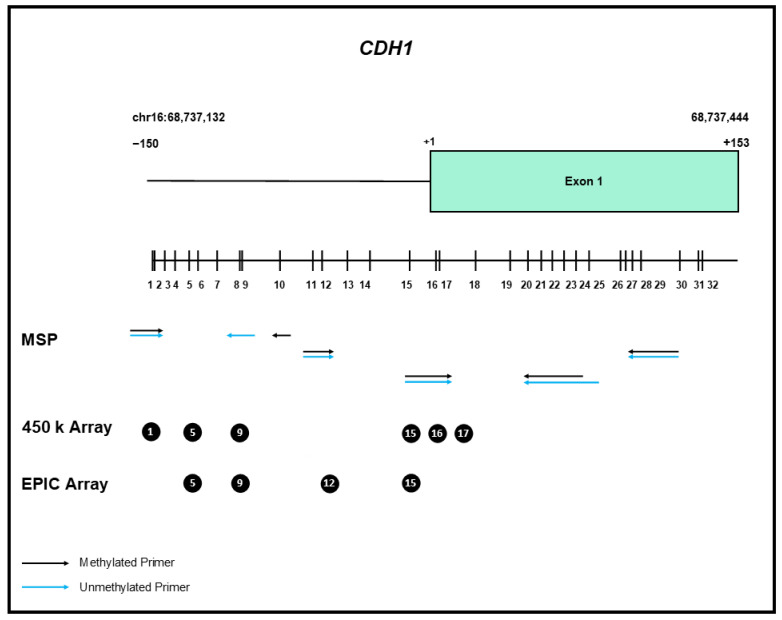
Schematic overview of the CpG islands encompassing the 5′ end of the *CDH1* gene. The individual CpG dinucleotides are represented by vertical bars, representing the distribution within the genomic sequence surrounding the start point of transcription (+1). The position of the primers for various MSP and qMSP assays are indicated by arrows, the CpG dinucleotides represented on the 450 k and EPIC array are indicated by circles.

**Table 1 cancers-14-04377-t001:** Detailed overview of the studies reporting CDH1 gene methylation in lobular breast cancer. Only publications analyzing more than 10 cases of lobular breast cancer are included.

Study	*n* =	Methylated Cases	Method	Reference
Droufakou et al. Int. J. Cancer 2001	22	17 (77%)	MSP	[75]
Sarrio et al. Int. J. Cancer	46	19 (41%)	MSP	[89]
Lombaerts et al. BBRC	11	8 (73%)	MSP	[90]
Shinozaki et al. Clin. Cancer Res. 2005	33	12 (36%)	MSP	[73]
Zou et al. J. Pathol. 2009	14	13 (93%)	MSP	[91]
Ciriello et al. Cell 2015	127	0	450 k array + NGS	[92]
Liu et al. Oncol. Lett. 2016	31	8 (26%)	MSP	[81]
Alexander et al. Cancers 2022	18	0	EPIC array	[93]

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
