# Peer review of "CDH1* (*E-cadherin*) Gene Methylation in Human Breast Cancer: Critical Appraisal of a Long and Twisted Story"

_cancers, 2022, doi:10.3390/cancers14184377_

Round 1
Reviewer 1 Report
This review provided updated comprehensive information on the role of E-Cadherin in breast cancer. The article is fit for publication in cancer. I would suggest improving the vitalization and resolution of Figures 1 and 2 prior to publication.
Author Response
Reviewer#1
1. “suggest improving the vitalization and resolution of Figures 1 and 2 prior to publication.”
Done as suggested

Reviewer 2 Report
The authors lay out a critique of not only the methodology used to investigate DNA methylation patterns in E-cadherin (CDH1), but also the conclusions made in review papers as a result of years of reporting data that likely represents false positive results. The implications are that much effort and resources are being misdirected in evaluating DNA methylation (not just for one gene in one tumor type, but across the board in cancer research) by inadequate methodology. The authors call for a standardization in the field of sound quantitative methods with adequate controls.
The arguments in this paper are logical and drawn from a large number of published studies. There are only a few minor changes I recommend below:
Line 33: Should be “methylation” instead of “ethylation”
Figure 1: “MethyQESD” instead of “MethylQESD” (Also, the text resolution can be enhanced in this figure.)
Line 227: “extent” instead of “extant”
Figure 2: This figure is a bit unclear. Do the numbers indicate CpG islands? The figure seems to imply that they are evenly distributed throughout the 5’ end of the CDH1 gene. Are there supposed to be circles other than the black dots shown by 450K Array and EPIC Array?
Line 268: The authors state, “Unfortunately, the presented primary data are difficult to evaluate.” This seems like a subjective statement and needs to be elaborated.
Line 399: “extent” instead of “extant”
Line 415: I looked up Ref [90] (Kristiansen et al. 2013) and found that the paper did in fact incorrectly cite Ref 11 in one instance. However, I did not see a mistake in it referencing Ref 30, since the authors (Kristiansen et al.) were knowingly discussing a different cadherin protein. If there is an important point here to be made, it probably needs to be further justified. Otherwise, I think this sentence could be omitted without doing harm to the paper’s main arguments.
Line 416-419: This quotation needs a brief explanation as to why it is significant.
Author Response
Reviewer#2
1. Typos in line 33, line 227, line 399, and Figure 1.
Corrected
2. Text resolution in Figure 1 “can be enhanced”.
Done as suggested
3. “Figure 2 is a bit unclear.”
3.1 Do the numbers indicate CpG islands?
No, individual CpG dinucleotides. This is now explained in the revised figure legend.
3.2 The figure seems to imply that they are evenly distributed throughout the 5’ end of the CDH1 gene.
Reviewer#2 correctly points out that the positions of the individual CpG dinucleotides were not drawn to scale in the first version of the manuscript. In the revised version of Figure 2 the vertical bars symbolizing each a CpG dinucleotide are distributed according their position within the sequence of the CDH1 gene.
3.3 Are there supposed to be circles other than the black dots shown by 450K Array and EPIC Array?
No, only the indicated 6 or 5 CpG dinucleotides are represented on the 450k and EPIC array, respectively. This is now explained in the revised figure legend.
4. Line 268: “seems like a subjective statement and needs to be elaborated.”
Reviewer#2 correctly points out that “Unfortunately, the presented primary data are difficult to evaluate.” is not a very precise scientific statement. Actually, no primary data for CDH1 are provided (only for GSTP1 and RAR-2). This is now mentioned in a new sentence (and the original sentence in line 276/268 is deleted).
5. Line 415: “I think this sentence could be omitted without doing harm to the paper’s main arguments.”
Reviewer#2 correctly points out, that Kristiansen et al. discuss on p. 142 of their review (right column) the methylation of CDH13, citing ref. 30 correctly (Toyooka et al., 2001).
However, in Table 1 Toyooka et al. (= ref. 30) is listed for CDH1. We came across this small mistake because we checked all publications listed in Table 1 for providing data about CDH1 gene methylation. Therefore, we left the manuscript unchanged at this position.
It is just another small piece of evidence that, unfortunately, data, claims, and references deposited in PubMed cannot be taken for granted without any further control. This is an important reminder, because in my experience many students are surprised and puzzled to find out that there are so many false claims and mistakes in PubMed.
5. Line 416-419: “This quotation needs a brief explanation as to why it is significant.”
Reference 94 (a very recent review about ILBC) is a good example, that after 25 years of research the vanishing importance of CDH1 gene methylation in ILBC is documented now also in reviews and not only in primary studies. This is now explained in an addition sentence in the revised version of the manuscript.

Reviewer 3 Report
In “CDH1 (E-Cadherin) Gene Methylation in Human Breast Cancer: Critical Appraisal of a Long and Twisted Story” the authors present the history of analysis of CDH1 methylation in breast cancers and how differences in experimental methodologies have led to disparate conclusions. The review is well-written and presents a cautionary tale about how “standard”, “accepted” methodologies can inadvertently lead to errors that can influence a field for years. This review was educational both in presenting the limitations of, and potential sources of error, for common methods for analyzing gene methylation and in how those errors impacted our understanding of whether methylation of the CDH1 gene was important in breast cancer.
Line 273, could be more clear.
Author Response
Reviewer#3
1. “Line 273, could be more clear.”
Following the suggestion of reviewer#3 the sentence in line 272 -274 was rephrased:
These authors also described CDH1 hypermethylation in six cases which were clearly positive for E cadherin expression in the tumor cells, demonstrated by unequivocal membranous E-cadherin staining (similar to Droufakou et al. ([68], see below).
